# Influence of Maternal BLV Infection on miRNA and tRF Expression in Calves

**DOI:** 10.3390/pathogens12111312

**Published:** 2023-11-03

**Authors:** Anna K. Goldkamp, Ciarra H. Lahuis, Darren E. Hagen, Tasia M. Taxis

**Affiliations:** 1Department of Animal and Food Sciences, Oklahoma State University, Stillwater, OK 74074, USA; anna.goldkamp@okstate.edu (A.K.G.);; 2Department of Animal Science, College of Agriculture and Natural Resources, Michigan State University, East Lansing, MI 48824, USA; lahuisci@msu.edu

**Keywords:** cattle, BLV, bovine leukemia virus, microRNAs, tRNA-derived fragments

## Abstract

Small non-coding RNAs, such as microRNAs (miRNA) and tRNA-derived fragments (tRF), are known to be involved in post-transcriptional gene regulation. Research has provided evidence that small RNAs may influence immune development in calves. Bovine leukosis is a disease in cattle caused by Bovine Leukemia Virus (BLV) that leads to increased susceptibility to opportunistic pathogens. No research has addressed the potential influence that a maternal BLV infection may have on gene regulation through the differential expression of miRNAs or tRFs in progeny. Blood samples from 14-day old Holstein calves born to BLV-infected dams were collected. Antibodies for BLV were assessed using ELISA and levels of BLV provirus were assessed using qPCR. Total RNA was extracted from whole blood samples for small RNA sequencing. Five miRNAs (bta-miR-1, bta-miR-206, bta-miR-133a, bta-miR-133b, and bta-miR-2450d) and five tRFs (tRF-36-8JZ8RN58X2NF79E, tRF-20-0PF05B2I, tRF-27-W4R951KHZKK, tRF-22-S3M8309NF, and tRF-26-M87SFR2W9J0) were dysregulated in calves born to BLV-infected dams. The miRNAs appear to be involved in the gene regulation of immunological responses and muscle development. The tRF subtypes and parental tRNA profiles in calves born to infected dams appear to be consistent with previous publications in adult cattle with BLV infection. These findings offer insight into how maternal BLV infection status may impact the development of offspring.

## 1. Introduction

In cattle, environmental impacts on dam health while pregnant can impact the progeny’s productive life. For example, heat stress in pregnant dams has been found to decrease lifespan and milk yield in both daughters and granddaughters [1]. Further, variations in maternal nutrition during pregnancy causes reduced immunity and an increased risk of obesity in progeny [2,3]. Arsenault et al. found that pups born to dams infected with a bacterial endotoxin lipopolysaccharide (LPS) or poly(I:C), a synthetic analog of viral dsRNA used to mimic the pathology of viral infection, experienced decreased growth rates and reduced immune signaling up to ten days of life [4]. Given the importance that early calf health has on productive potential, it is important to consider the effects of maternal environment on progeny development. 

Epigenetic studies may aid to further investigate the influences of the intrauterine environment on progeny development. MicroRNAs (miRNA) are a non-coding RNA species, ranging from 18 to 24 nucleotides in length, that play a role in epigenetic regulation through post transcriptional gene regulation [5]. MiRNAs function through the repression of target genes in various ways. Complementary binding to mRNA can either provoke cleavage or the translational repression of the targeted mRNA [6]. Research also suggests miRNAs can cause deadenylation of a target mRNA, leading to instability and repression of the targeted mRNA [7]. In addition, tRNA-derived fragments (tRFs) are a relatively new class of small non-coding RNAs which are formed through the cleavage of mature tRNA molecules [8,9]. Different subtypes of tRFs may be produced based on the position in which the mature tRNA is cleaved. In addition, 5′ tRFs, 3′ tRFs, and internal-tRFs (i-tRFs; tRFs derived from within the mature tRNA) range from 16 to 26 nucleotides, whereas 5′ and 3′ tRNA halves range from 27 to 36 nucleotides [10,11]. Recent research suggests that tRFs function similarly to miRNAs, in which they are recruited to an RNA-induced silencing complex to target complementary mRNA for degradation [12]. Furthermore, the literature demonstrates that placental small non-coding RNAs, such as miRNAs, can move through maternal circulation and be trafficked to the placenta and fetal compartment, and vice versa, placental RNAs can be transmitted to maternal plasma [13,14,15]. While there is existing evidence demonstrating a relationship between miRNA, tRFs, and Bovine Leukemia Virus (BLV), the literature exploring the potential effects of dam BLV infection on its offspring is nonexistent [16,17,18].

Bovine Leukemia Virus is a delta retrovirus infecting cattle and the etiologic agent causing enzootic bovine leukosis [19]. Infection by BLV presents in several stages within the host. About 60 to 70% of BLV-infected animals remain asymptomatic, 30% develop persistent lymphocytosis, and 2 to 5% develop malignant lymphoma [20,21]. Dairy producers may suffer increased costs due to BLV infection in the herd. For example, BLV-infected animals that develop persistent lymphocytosis are at an increased risk for also developing B lymphocyte lymphoma, which is related to a weakened immune system and may lead to increased susceptibility to infection by other opportunistic pathogens [21,22,23]. Additionally, malignant lymphoma leads to carcass condemnation at slaughter, a direct profit loss to producers [24,25]. Infection by BLV can be found by a quantification of proviral load (PVL) tested by a qPCR assay, where a greater PVL is indicative of disease progression [26]. However, it is worth acknowledging that serological tests, including Agar gel immunodiffusion (AGID) and enzyme-linked immunosorbent assay (ELISA), are recommended by the World Organization for Animal Health (WOAH) in order to detect BLV infection [27].

In 2008, the United States Department of Agriculture (USDA) estimated 89% of US dairy operations had at least one BLV seropositive animal in their herd [28]. Because BLV is a bloodborne pathogen, cattle residing on farms with at least one BLV-infected animal are at risk for spreading the virus by blood transfer via the reuse of examination sleeves, hoof trimmings, biting flies, hypodermic needles, and dehorning tools [29,30,31].

While research continues to explore the effects of BLV on dairy cattle and the dairy industry, studies exploring the potential effects that dam BLV infection has on the early development of progeny is lacking. The present study aims to identify differences in miRNA and tRF expression in calves born to BLV-infected dams. These results provide evidence of potential alterations in development in calves born to BLV-infected dams. Specifically, the differential expression of miRNAs involved in the gene regulation of the immunological response, cardiac development, and skeletal muscle development of calves born to BLV-infected dams was observed. In addition, novel tRFs specific to BLV infection were identified.

## 2. Materials and Methods

### 2.1. Samples

The use of all animals in this study was approved by the Institutional Animal Care and Use Committee. All study dams were enrolled in a larger study, which required supplementation of choline at pre-parturition for approximately 30 days. Treated dams were supplemented with 30 g (n = 10) or 45 g (n = 5) of choline per day, and control dams were not fed choline (n = 7). All study calves were purebred Holstein calves. Twelve treatment calves were born to BLV PVL-positive dams, and ten control calves were born to BLV PVL-negative dams (Table 1; Appendix A). All calves were fed Bovine IgG Colostrum 200 (Saskatoon Colostrum Company, Saskatoon, SK, Canada) to supplement maternal colostrum, followed by Calf’s Choice Total HiCal (Saskatoon Colostrum Company, Saskatoon, SK, Canada) throughout the study period of three weeks. 

Two blood samples were collected via jugular venipuncture from all calves into a PAXgene Blood RNA tube (Becton Dickinson, Franklin Lakes, NJ, USA) and a vacutainer K2 EDTA tube (Becton Dickinson, Franklin Lakes, NJ, USA) at 14 days of age. Samples in PAXgene blood RNA tubes were stored at 4 °C for 16 to 24 h, then stored at −80 °C for further analysis. Blood stored in vacutainer K2 EDTA tubes was immediately used to determine lymphocyte count (LC) via the QScout (Advanced Animal Diagnostics), then transferred to the laboratory for white blood cell (WBC) preservation.

### 2.2. White Blood Cell Preservation

In brief, red blood cell lysis (ThermoFisher Scientific, Waltham, MA, USA) was diluted to 1× concentration and 1.5 mL was added to microcentrifuge tubes containing 500 μL of whole blood. All aliquots were inverted for 10 min at 20 °C, then centrifuged at 500× *g* for 5 min at 20 °C. The top plasma layer and lysed red blood cells were removed, avoiding the WBC pellet. The described steps were repeated once more to achieve optimal purification. Finally, 500 μL of RNAlater (ThermoFisher Scientific, Waltham, MA, USA) was added to each sample, and samples were incubated overnight at 4 °C before long term storage at −80 °C. Preserved WBCs were used for PVL determination.

### 2.3. BLV Antibody and BLV PVL Determination

Gp51 antibody capture ELISA test kit (IDEXX Laboratories, Inc., Westbrook ME, USA) was used to determine anti-BLV antibodies from calf plasma samples, as previously described by Hutchinson et al., 2020 [32] (Table 1). Additionally, both colostrum supplements were assayed for anti-BLV antibodies via milk ELISA. 

To determine BLV PVL, DNA extraction was performed from frozen WBC lysates of all 14-day-old calf samples using the DNeasy Blood and Tissue Kit (Qiagen, Hilden, Germany). To determine BLV PVL for both dam and calf samples, the SS1 qPCR assay (CentralStar Cooperative, Lansing, MI, USA) was used as previously described [33]. In short, the SS1 qPCR assay is a multiplex qPCR assay with primers and probes designed to target the BLV polymerase gene, bovine ß-Actin, and a spike-in control oligo allowing for quantification of PVL. To achieve a PVL, a standard curve was prepared by using linearized plasmids quantified by droplet digital PCR. The copy numbers for both BLV and bovine ß-Actin were obtained from the previously prepared standard curve. The final value for PVL was expressed as a ratio of BLV copies to bovine ß-Actin copies [33]. Previous analysis of PVL using the SS1 assay shows an analytical sensitivity of as little as 0.001 copies per cell, which equates to 1 copy per 1000 cells [32,34].

### 2.4. RNA Extraction, Library Preparation, and Sequencing

RNA was isolated from stored PAXgene blood RNA tubes by first centrifuging all tubes at 3000× *g* for 10 min, decanting the supernatant, and adding 4 mL of RNase free water. The pellet was vortexed to dissolve into solution, and the centrifugation step was repeated. The supernatant was decanted once again. Total RNA was extracted using the mirVana miRNA Isolation kit following manufacturer’s protocol (ThermoFisher Scientific, Waltham, MA, USA). Small RNA libraries were prepared using the NEBNext Small RNA Library prep kit (New England Biolabs, Ipswich, MA, USA) following manufacturer’s protocol. RNA quality and quantity was assessed using the Agilent 2100 Bioanalyzer (Agilent, Santa Clara, CA, USA). The 22 pooled samples were sequenced using a single S1 flow cell on the Illumina NovaSeq 6000 (Illumina, San Diego, CA, USA). Approximately 37 million single-end reads of 100 nucleotides were obtained for each sample.

### 2.5. Small RNA Processing

Adapter sequence (AGATCGGAAGAGCACACGTCTGAACTCCAGTCAC) was removed from the raw sequencing reads by using cutadapt v1.18. Reads with low quality bases (Phred score < 20) were trimmed using the dynamictrim function of SolexaQA++ v3.1.7.1 [35]. Reads with lengths shorter than 17 bases were discarded after adapter and quality trimming, and the remaining reads were aligned to the bovine genome (ARS-UCD1.2) [36] using miRDeep2 v0.0.8 [37]. The parameter allowing a configuration file was used to allow all files to be processed together. Reads aligned to known bovine miRNAs from miRBase (release 22 [38]) were quantified using the default settings of miRDeep2 in which only one mismatch was allowed within the read. In addition, only reads that fit the specified default setting region, namely the length of the miRNA (≥17–≤25 nucleotides), and 7 nucleotide flanking regions (i.e., ≤2 nucleotides upstream and ≤5 nucleotides downstream of the miRNA) were counted. Reads mapped to miRNAs with multiple precursors were only counted once for each read per miRNA. 

For tRF prediction, the MINTmap pipeline was utilized. Briefly, genomic tRNA sequences were retrieved from gtrnadb (http://gtrnadb.ucsc.edu) and mitotRNAdb (http://mttrna.bioinf.uni-leipzig.de) accessed on 1 August 2022. Custom scripts were used to remove introns, allow discriminator bases at the −1 position, and add CCA tails. A sliding window was used to break each sequence into lengths ranging from 16 to 50 nucleotides to generate a list of all possible tRF sequences. To determine exclusivity, a masked genome file was created marking tRNA exons with a “1”, post-transcriptional modifications (−1 discriminator and CCA tail) with a “2”, and everything else with a “0”. Processed reads were then aligned to the candidate tRF sequences determined as exclusive or ambiguous to the tRNA space. Exclusive tRFs are those that only map to annotated tRNAs and nowhere else in the genome. Only exclusive tRFs were kept for the analysis.

### 2.6. Differential Expression Analysis

The betweenLaneNormalization function of EDAseq v2.24.0 was used for full-quartile normalization and the plotPCA function of the DESeq2 package was used for principal component analysis visualization [39]. Differential expression analysis was performed with EdgeR v3.32.1 [40]. Raw counts were first normalized with the Trimmed Mean of M values (TMM) parameter within the calcNormFactors function. Dispersions were estimated based on normalized counts using estimateDisp function and a likelihood ratio test was performed with functions glmFit and glmLRT of edgeR to identify differentially expressed miRNAs and tRFs [38]. Three differential expression tests were performed: (1) calves born to BLV positive dams vs. calves born to BLV negative dams; (2) calves born to BLV positive choline-control dams vs. BLV positive choline-30 g/day dams; (3) calves born to BLV positive choline-control dams vs. BLV positive choline-45 g/day dams. MicroRNAs and tRFs with a *p*-value and false discovery rate ≤ 0.05 were considered significant (Table 2).

## 3. Results and Discussion

All colostrum supplements and calves were positive for the presence of anti-BLV antibodies via ELISA (Table 1). Samples from the calves were all BLV PVL-negative, indicating no detectible levels of BLV provirus. Therefore, it is probable that the calves obtained anti-BLV antibodies from the consumption of colostrum supplements. Although it is unlikely given the age that the calves were sampled, we acknowledge that BLV provirus levels may not have been high enough for detection. 

To evaluate the relationship between BLV and the profiles of miRNAs and tRFs, small RNA sequencing was performed on total RNA isolated from the blood of calves born to dams testing positive or negative for BLV. Using miRDeep2 and MINTmap, the expression of 837 miRNAs and 17,107 predicted tRFs was detected across all samples (Appendix A). A principal component analysis (PCA) showed variation in small RNA expression among positive and negative samples (Figure 1). More specifically, PC1 described 16.5% and 11.11% of the variation in miRNA and tRF expression, respectively, whereas PC2 described only 15.24% and 8.53% of variance in miRNA and tRF expression. This indicates that miRNA and tRF expression only explains a small amount of variation to differentiate between progeny born to BLV-infected dams and BLV negative dams, and small RNA expression may also be influenced by other factors. In order to also evaluate the impacts of choline treatment on innate immunity, a subset of dams in our study were supplemented with choline at pre-parturition for approximately 30 days. When comparing BLV positive progeny without choline treatment to BLV positive progeny born to dams supplemented with 30 g of choline per day, there were no differentially expressed miRNAs or tRFs. However, we observed that 45 g of choline per day specifically impacted the expression of tRFs, where we identified 34 differentially expressed tRFs between control BLV progeny and BLV progeny born to dams supplemented with 45 g of choline per day (Appendix A). Although no alterations in miRNA expression due to choline supplementation were found, the detection of differentially expressed tRFs due to 45 g of choline supplementation suggests that choline use influences gene expression. Indeed, maternal choline supplementation has been suggested to modify the epigenetic programming of offspring perhaps resulting in improved health outcomes [41]. While further investigation is necessary to identify tRF gene targets and downstream responses due to the supplementation of choline, these findings could aid in the development of a therapeutic approach to modulate epigenetic mechanisms for BLV treatment.

We identified five miRNAs (bta-miR-1, bta-miR-206, bta-miR-133a, bta-miR-133b, and bta-miR-2450d) with altered expression in calves born to BLV-infected dams compared with those born to BLV negative dams (Table 2). A previous study found that bta-miR-206 and bta-miR-133a were downregulated in the serum of BLV positive cows, and also predicted BLV regulatory genes, *rex* and *tax*, to be targets of bta-miR-206 [16]. In our study, bta-miR-206 and bta-miR-133a were downregulated in calves born to BLV positive cows, which suggests that BLV infection in dams can have repercussions in their progeny. Specifically, this could indicate that maternal dysregulation of miRNA expression may be transmitted to progeny. Another publication reported that miR-133b and miR-206 are coregulated with genes responsible for IL-17 production in circulating human and mouse T-cells [42]. IL-17 is a cytokine which fights extracellular bacteria and fungi during an immune response. This implies that the downregulation of bta-miR-133b and bta-miR-206 may be associated with a decrease in IL-17 expression or an immunological response. Calves are often infected with *E. coli* on dairy operations, leading to diarrhea and even death [43,44]. An impaired immunological response to fungi or bacterial infections in calves born to BLV infected dams may lead to a poor outcome for calves infected with bacteria, such as *E. coli*. 

Human and mouse orthologs of the four downregulated miRNAs (miR-1, miR-133a, miR-133b, and miR-206) have been shown to play roles in cardiac and skeletal muscle growth and differentiation [45,46,47]. Specifically, miR-1 targets the Hand2 transcription factor as well as HDAC4, which are necessary for cardiac development and muscle growth repression, respectively [48,49]. In addition, miR-133 seems to oppose the function of miR-1 by targeting the SRF gene, which functions to activate muscle growth [50]. Further, miR-206 indirectly upregulates MyoD, leading to muscle cell differentiation [51]. These findings suggest that the presence of BLV infection in dams may influence skeletal muscle and cardiac growth and development in their progeny. One miRNA could have the potential to regulate several genes and similarly one gene could be regulated by multiple miRNAs [52]. This is due to their ability to regulate expression based on partial or full complementarity to the transcript sequence [53,54].

Emerging evidence suggests that tRFs can regulate cellular and molecular mechanisms contributing to cell states, tissue types or disease development [55,56,57]. In 2009, Lee et al. discovered the novel class of small non-coding RNAs, which exhibited specific expression profiles and participated in vast biological processes [10]. Given that research on tRFs is a rather new field that only garnered recognition about a decade ago, few studies have investigated their relationship with BLV. Therefore, we also investigated dysregulated tRF expression between calves born to BLV positive dams and calves born to BLV negative dams. Overall, we identified 2 downregulated (tRF-36-8JZ8RN58X2NF79E and tRF-20-0PF05B2I) and 3 upregulated (tRF-27-W4R951KHZKK, tRF-22-S3M8309NF, and tRF-26-M87SFR2W9J0) tRFs in calves born to BLV positive dams (Table 2). Of these 5 differentially expressed tRFs identified, 4 were i-tRFs and 1 was a 5′ tRF (tRF-22-S3M8309NF). As the MINTmap pipeline also provides the parental tRNA source from which each tRF is derived, we found that 2 of the differentially expressed tRFs (tRF-27-W4R951KHZKK and tRF-26-M87SFR2W9J0) were derived from tRNAs for LysCTT, whereas the other tRFs (tRF-36-8JZ8RN58X2NF79E, tRF-20-0PF05B2I, and tRF-22-S3M8309NF) were derived from Mt-ProTGG, Mt-ThrTGT, SerAGA, respectively. Because tRF abundance is associated with tRNA expression, future work should investigate variations in mature tRNA expression due to BLV status.

In order to comprehensively investigate tRFs, expression profiles of all expressed tRFs by subtype were examined (Figure 2). Notably, the majority of expressed tRFs belonged to the 5′ tRF subtype in progeny born to BLV negative and positive dams. Our findings are consistent with a previous study, where tRF expression in white blood cells of Holstein cattle infected with BLV showed that the majority of tRNA fragments were derived from the 5′ end of mature tRNA sequences (5′ tRF) [17]. We also elucidated the parental tRNA contribution to the predicted pool of tRFs (Figure 3). Overall, we found that 62.54% and 37.46% of all expressed tRFs were derived from nuclear and mitochondrial tRNAs, respectively. Moreover, within these categories, the majority of mitochondrial tRFs were derived from SerUGA, while most nuclear tRFs originated from HisGUG in BLV negative and positive groups (Figure 3). Similar to a previous BLV study, we also found that Histidine, Glutamine, and Glycine exhibit substantial contributions to tRF production [17]. These results suggest that tRF expression profiles in calves born to BLV infected dams may be maternally inherited due to their resemblance to the profiles of infected dams. This may suggest selective transcription of specific tRNA species to possibly modulate the tRF profile in order to effectively regulate gene expression in calves responding to BLV infection.

## 4. Conclusions

Previous research has identified the impact that a dam’s viral infection status has on the progeny’s growth and immunological development [4]. Other environmental factors affecting the dam, such as heat stress and nutrition, have been associated an with impaired productive life of the progeny [1,2,3]. The mechanisms by which the intrauterine environment may impact the health of progeny can be studied through identifying epigenetic regulation patterns including investigating DNA methylation patterns, histone modification, or small non-coding RNA expression. The present study identifies differences in miRNA and tRF expression in progeny born to BLV-infected dams in an effort to obtain evidence of the potential misregulation of genes in the calves. 

MicroRNAs have been shown to play a role in early calf development [46,47]. In the current study, differentially expressed miRNAs were found to be dysregulated in calves born to BLV-infected dams. This work revealed that bta-miR-206 and bta-miR133a is downregulated in calves born to BLV positive dams, which coincides with a previous study that these miRNAs are also downregulated in cattle with active BLV infection [16]. These findings suggest that in utero exposures, such as BLV infection, can impact miRNA expression and health outcomes in offspring via RNA interference. In addition, miR-133b and miR-206 were downregulated in progeny born to BLV positive dams and have been found to coregulate with IL-17 production in T-cells [42]. 

We have also presented early evidence that tRF expression can be influenced by maternal BLV infection. In total, we identified five differentially expressed tRFs originating from mitochondrial and nuclear tRNAs. By exploring the subtypes and parental tRNAs of all expressed tRFs, we identified that 5′ tRFs are the most predominant subtype, which was also supported in a study assessing tRF expression in BLV-infected mature Holstein cows [17]. Furthermore, our work and the aforementioned study demonstrated that the majority of tRFs are derived from parental tRNAs coding for Histidine, Glutamine, and Glycine. These results could suggest altered mature tRNA expression for the modulation of translation or the production of regulatory products.

Overall, both miRNA and tRF expression can fluctuate depending on stage of life, immune regulations, disease, and other biological processes within an organism [16,17,58,59,60,61]. Since samples were collected at a single time point (14 days of age), temporal studies should be completed to assess the differential expression of small non-coding RNAs in calves to achieve a clear understanding of the epigenetic effects a BLV-infected dam plays on a calf’s post-natal development. It may also be valuable to collect animal metadata on disease status and growth throughout life to further investigate the role miRNAs and tRFs play in the development and productive life of progeny.

## Figures and Tables

**Figure 1 pathogens-12-01312-f001:**
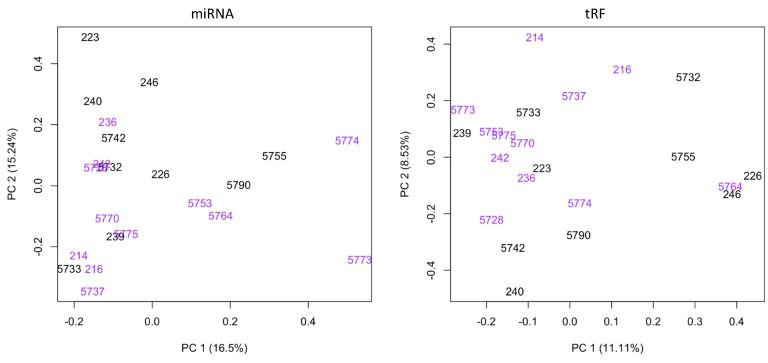
Principal component analysis (PCA) shows variation in miRNA and tRF expression between progeny born to BLV negative (black text) and BLV positive (purple text) animals. The x-axis represents the first principal component (PC1), which shows the individuals with the most variability with regards to miRNA and tRF expression. The y-axis represents the second principal component (PC2), which reveals additional relationships in the data that are not explained by PC1. PC1 and PC2 represent linear combinations based on the gene expression data and are used to calculate maximum variance in the dataset. PC1 explains the most variation and PC2 explains the second most variation.

**Figure 2 pathogens-12-01312-f002:**
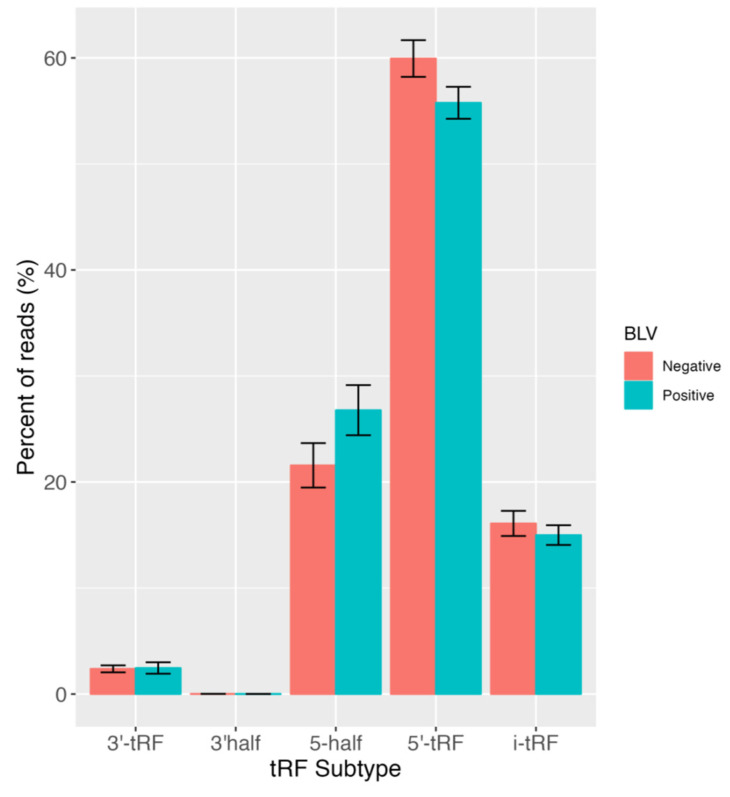
Distribution of tRF subtypes between progeny born to BLV positive and BLV negative dams. Bar graph showing the percentage of reads assigned to 5′ tRF, 3′ tRF, i-tRF, 5′ half and 3′ half subtypes. All reads were counts per million (CPM) normalized and percentages contributing to the total tRF profile were calculated. The y-axis sums to 100% for each BLV status and standard error bars are shown in black.

**Figure 3 pathogens-12-01312-f003:**
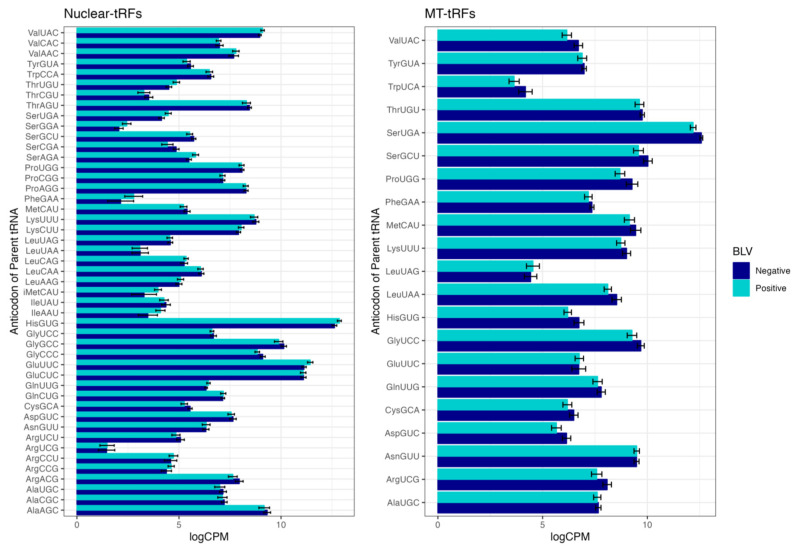
Nuclear and Mitochondrial tRF distribution between progeny born to BLV negative and BLV positive dams. Reads were logCPM normalized and the number of reads derived from each parent tRNA are shown. The SummarySE function of the Rmisc package was used to calculate summary statistics by treatment group and the standard error bars for each parent tRNA within each treatment group are shown.

**Table 1 pathogens-12-01312-t001:** Summary Data of Study Animals.

Calf ID	Sex ^1^	Calf ELISA OD ^2^	Dam BLV Status ^3^
1	M	2.507	Negative
2	F	2.253	Negative
3	M	2.441	Negative
4	F	2.585	Negative
5	F	1.246	Negative
6	M	2.461	Negative
7	F	1.952	Negative
8	F	2.257	Negative
9	M	2.756	Negative
10	M	2.765	Negative
11	F	2.061	Positive
12	F	2.622	Positive
13	M	3.093	Positive
14	M	1.611	Positive
15	F	2.720	Positive
16	F	2.820	Positive
17	F	2.596	Positive
18	F	2.325	Positive
19	M	2.232	Positive
20	F	1.927	Positive
21	F	2.052	Positive
22	M	2.909	Positive

^1^ M or F describing a male or female calf enrolled in the study. ^2^ ELISA optical density describing the optical density exhibited by anti-BLV antibodies. ^3^ Dam BLV status as determined by SS1 qPCR assay.

**Table 2 pathogens-12-01312-t002:** Differential Expression of miRNAs and tRFs in calves born to BLV Infected Dams.

miRNAs/tRFs	logFC ^1^	logCPM ^2^	LR ^3^	*p*-Value	FDR ^4^
bta-miR-1	−6.835	5.777	31.975	1.56 × 10^−8^	1.61 × 10^−5^
bta-miR-206	−6.072	4.258	28.535	9.20 × 10^−8^	4.74 × 10^−5^
bta-miR-133a	−4.401	3.196	20.003	7.73 × 10^−6^	2.65 × 10^−3^
bta-miR-133b	−4.653	0.494	16.522	4.81 × 10^−5^	1.24 × 10^−2^
bta-miR-2450d	4.102	−3.528	14.012	1.82 × 10^−4^	3.74 × 10^−2^
tRF-27-W4R951KHZKK	2.874	8.612	27.967	1.23 × 10^−7^	2.11 × 10^−3^
tRF-36-8JZ8RN58X2NF79E	−4.028	4.762	20.405	6.27 × 10^−6^	4.81 × 10^−2^
tRF-20-0PF05B2I	−3.743	4.868	19.300	1.12 × 10^−5^	4.81 × 10^−2^
tRF-22-S3M8309NF	4.074	5.061	19.040	1.28 × 10^−5^	4.81 × 10^−2^
tRF-26-M87SFR2W9J0	3.269	5.613	18.861	1.41 × 10^−5^	4.81 × 10^−2^

^1^ log fold change. ^2^ log counts per million. ^3^ likelihood ratio. ^4^ false discovery rate.

## Data Availability

Small RNA sequencing data reads are deposited in FASTQ format to the NCBI Sequence Read Archive database (SRA) under the Bioproject accession number PRJNA1023946.

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
