# Peer review of "Influence of Maternal BLV Infection on miRNA and tRF Expression in Calves"

_pathogens, 2023, doi:10.3390/pathogens12111312_

Round 1

Reviewer 1 Report

Comments and Suggestions for Authors

The reviewed manuscript (“ Influence of Maternal BLV Infection on miRNA and tRF Expression in Calves”) is well written and deals with new issue – whether non-coding RNA (miRNA and tRF) present in maternal BLV infection may impact the development of offspring.

The reviewer has some minor observations:

1. Abstract, Line: 23-24 “ The tRF subtypes and parental tRNA profiles appear to be consistent with previous publications in cattle.” This sentence should be reconsidered, authors wrote they received the same results as before - so why did they write the paper? It is not very clear what result of previous research the authors are talking about.

2. Introduction, Line 33-34 “Arsenault et al. found that pups born to virally infected dams experienced decreased growth rates and reduced immune signaling up to ten days of life.” The phrase " virally infected dams " does not clearly specify what virus the authors are talking about.

3. Introduction, Line 44-45 the sentence : “ In addition, tRNA-derived fragments (tRFs) are a relatively new class of 44 small non-coding RNAs which are formed through cleavage of mature tRNA molecules” does not have a reference assigned to it.

4. Introduction, Line: 51-52: “While there is existing evidence demonstrating a relationship between miRNA, tRFs, and Bovine Leukemia Virus (BLV), literature exploring the potential effects of dam BLV infection on its offspring is nonexistent.” Please provide references with evidence of an association with BLV or Retroviruses.

5. Introduction, Line 57: “Dairy producers may suffer increased costs and decreased profit due to BLV infection in the herd.” It's either increased costs or decreased profit - it's the same thing.

6. Introduction, Line 60 :” „In fected animals that develop persistent lymphocytosis are susceptible to infection by other opportunistic pathogens, increasing vet costs for the producer [13].” Pathogens and opportunistic diseases are not listed in this cited paper ("Herd-level determinants of bovine leukaemia virus prevalence in dairy farms”).

7. Table 1. Calf ELISA OD column. “ELISA optical density describing the optical density exhibited by anti-BLV antibodies.” The reported OD values do not differ between the groups of BLV-infected and uninfected cows. If they determine anti-BLV antibodies, they should differ between groups.

8. 2.3. BLV antibody and BLV PVL Determination. Why did the authors use plasma instead of serum for the ELISA test?

9. The description under the figure 1 should say PCA instead of PCR.  Figure 1. Principal component analysis (PCR) shows variation in miRNA…”

10. Figure 1. The description under the drawing is incomplete: 1) what the numbers on the chart mean; 2) no discussion of what the drawing shows; 3) why are such low percentages as 8.5% and 11..% found for PC1 and PC2? 4) what does this mean?

11. Results and Discussion, Line 187-188. Figure 1 is not described.

12. Results and Discussion, Line 195-197. “Although no alterations in miRNA expression 195 were illustrated in these comparisons, differential expression due to choline treatment 196 could suggest that choline use influences gene expression “ . This sentence is incomprehensible - what does "differential expression" refer to in this sentence? the authors wrote at the beginning of the sentence that no changes in miRNA expression were observed.

13. Results and Discussion, Line 228-229. “Emerging evidence suggests that tRFs can describe cellular and molecular mechanisms contributing to cell states, tissue types or disease development [33-35].” It would be better to say ‘regulate’ instead of ‘describe’.

14. Results and Discussion, Line 228-229. “Given that tRFs are a progressing category of small non-coding RNAs, few studies have investigated 230 their relationship with BLV.” How do the authors understand the statement 'progressing category' - for a reader who is just exploring the subject of tRF/miRNA/BLV, this formulation is not understandable. Additionally, the subject of tRF/BLV is relatively poorly understood.

15. In lines 211-213 it was described that miR-133b and miR-206 orthologs were associated with IL-17 production, while in lines 219-220 the same orthologies were associated with "cardiac and skeletal muscle growth and differentiation, influence skeletal muscle and cardiac "growth". Is this how it should be understood that one and the same orthologs - miR-133b and miR-206 in humans and mice have many functions? Line 211-213 “Another publication reported that miR-133b and miR-206 are coregulated with genes responsible for IL-17 production in circulating human and mouse T-cells [23]. IL-17 is a cytokine which fights extracellular bacteria and fungi during an immune response.

Line 219-220: “Human and mouse orthologs of the four downregulated miRNAs (miR-1, miR-133a, miR-133b, and miR-206) have been shown to play roles in cardiac and skeletal muscle growth and differentiation”

Overall though, this is a timely and needed paper. It is well researched and nicely written. With modifications addressing the detailed comments above this will be a worthwhile research paper.

Comments on the Quality of English Language

In my suggestions for authors, I list places in the manuscript that require improvement in English

Author Response

Hello and thank you for your comprehensive review of our work and manuscript. We have addressed your comments within the manuscript as well as in the attached document(s). We are hopeful these edits will address your concerns and lead to a successful publication. Thank you again for your time. 

Reviewer 2 Report

Comments and Suggestions for Authors

In this manuscript the authors revealed the differential expression of miRNAs and tRFs between calves born to BLV infected and uninfected dams. The study is of interest for both BLV biology and bovine immunology. My comments to improve the manuscript are shown below.

 Major comments

1 The authors identified several miRNAs and tRFs that were differentially expressed between calves born to BLV infected and uninfected dams. Why and how these differences occurred? Are miRNAs and tRFs profiles of dams relevant to that of calves? How the dam BLV infection and miRNAs and tRFs expression profile influence calves miRNAs and tRFs profile? The authors refer to [4] and [5], but these papers are not about bovine, retrovirus, miRNAs or tRFs. The authors need to discuss detailed and potential mechanism of influence of maternal virus infection to calve miRNAs and tRFs profile. Further, to discuss biological meaning of the differential expression of miRNA and tRFs of calves, dam miRNA and tRFs need to be analyzed.

 2 Comparative analysis of dam proviral load and calf miRNAs and tRFs profile are informative. Immunological responses to BLV are not same between cattle with high proviral load and that with low proviral load. Therefore, the dams proviral loads may influence the calves’ miRNAs and tRFs profiles.

 Line73: Is there a correlation between the miRNA and tRFs profiles of calves and their body weight, length, and clinical condition? The influence of miRNA and tRFs profiles to these phenotypes are worth analyzing.

Minorcomments

Line 33-35: Suggest to change “virally infected dams” to “poly I:C exposed dams”

Line 117: The authors need to add the brief method, specificity, and sensitivity of SS1 qPCR.

Table S1: The authors need to add the explanation on the value of Dam_SS1 and Dam_OSI (viral copy number/cell?)

Line 286: “in utero exposures” None of calves were infected with BLV, but exposed in utero?

Reviewer 3 Report

Comments and Suggestions for Authors

The manuscript investigated the variations in miRNA and tRF expression in calves born to dams infected with bovine leukemia virus. This is a relevant contribution to the field of infection with BLV and the paper is within the scope and interests of the Pathogens. The objective of this manuscript is not really new as there were some publications describing miRNA profile in calves. However, the novelty  the paper is the evidence that calves born to dams infected with BLV showed differential expression of miRNAs involved in gene regulation of some physiological processes. Furthermore, the paper describes for the first time a new tRFs, specific to BLV infection. The manuscript is generally well written and results are clearly presented. Nevertheless, I identified some point that should be clarified.

Lines 62-63  - the reader may give a wrong impression that the detection of BLV infections is based on
performing a quantitative PCR reaction. In my opinion, it should be pointed out that serological methods
(AGID, ELISA) recommended by the WOAH are essential for the detection of infections.

Line 90 -  As can be seen from Table 1, the BLV status of the mothers was assessed exclusively by the qPCR technique. This is surprising, as the authors analysed the status of the calves using both ELISA and qPCR. It seems that the "Negative" status of the dams should also be confirmed by ELISA. This is due to the fact that many times in cattle naturally infected with BLV, a negative qPCR result is accompanied by the presence of specific antibodies.  

Author Response

(The authors gave the same response as above.)

Round 2

Reviewer 2 Report

Comments and Suggestions for Authors

The authors have addressed all comments in a satisfactory manner.